# ADAPTIVE RISK MINIMIZATION: A META-LEARNING APPROACH FOR TACKLING GROUP SHIFT

## ABSTRACT

A fundamental assumption of most machine learning algorithms is that the training and test data are drawn from the same underlying distribution. However, this assumption is violated in almost all practical applications: machine learning systems are regularly tested under *distribution shift*, due to temporal correlations, particular end users, or other factors. In this work, we consider the setting where the training data are structured into groups and test time shifts correspond to changes in the group distribution. Prior work has approached this problem by attempting to be robust to all possible test time distributions, which may degrade average performance. In contrast, we propose to use ideas from meta-learning to learn models that are *adaptable*, such that they can adapt to shift at test time using a batch of unlabeled test points. We acquire such models by learning to adapt to training batches sampled according to different distributions, which simulate structural shifts that may occur at test time. Our primary contribution is to introduce the framework of adaptive risk minimization (ARM), a formalization of this setting that lends itself to meta-learning. We develop meta-learning methods for solving the ARM problem, and compared to a variety of prior methods, these methods provide substantial gains on image classification problems in the presence of shift.

## 1 INTRODUCTION

The standard assumption in empirical risk minimization (ERM) is that the data distribution at test time will match the distribution at training time. When this assumption does not hold, the performance of standard ERM methods typically deteriorates rapidly, and this setting is commonly referred to as distribution or dataset *shift* (Quiñonero Candela et al., 2009; Lazer et al., 2014). For instance, we can imagine a handwriting classification system that, after training on a large database of past images, is deployed to specific end users. Some new users have peculiarities in their handwriting style, leading to shift in the input distribution. This test scenario must be carefully considered when building machine learning systems for real world applications.

Algorithms for handling distribution shift have been studied under a number of frameworks (Quiñonero Candela et al., 2009). Many of these frameworks aim for *zero shot generalization* to shift, which requires more restrictive but realistic assumptions. For example, one popular assumption is that the training data are provided in *groups* and that distributions at test time will represent either new group distributions or new groups altogether. This assumption is used by, e.g., group distributionally robust optimization (DRO) (Hu et al., 2018; Sagawa et al., 2020), robust federated learning (Mohri et al., 2019; Li et al., 2020), and domain generalization (Blanchard et al., 2011; Gulrajani & Lopez-Paz, 2020). Constructing training groups or tasks in practice is generally accomplished by using meta-data, which exists for most commonly used datasets. This assumption allows for more tractable optimization and still permits a wide range of realistic distribution shifts. However, achieving strong zero shot generalization in this setting is still a hard problem. For example, DRO methods, which focus on achieving maximal worst case performance, can often be overly pessimistic and learn models that do not perform well on the actual test distributions (Hu et al., 2018).

In this work, we take a different approach to combating group distribution shift by learning models that are able to deal with shift by *adapting* to the test time distribution. To do so, we assume that we can access a batch of *unlabeled* data points *at test time* – as opposed to individual isolated inputs – which can be used to implicitly infer the test distribution. This assumption is reasonable

Figure 1: A schematic of the ARM problem setting and approach, described in detail in Section 3. Left: During training, we assume access to labeled data along with group information $z$, which allows us to construct training distributions that exhibit group distribution shift. For example, a training distribution may place uniform mass on only a single user's examples. We use these training distributions to learn a model that is adaptable to distribution shift via a form of meta-learning. We detail the specific adaptation procedures (orange box) that we consider in Section 3 and Figure 2. Right: We perform unsupervised adaptation to different test distributions, without requiring zero shot generalization to shift as in prior methods. If the test shifts we observe are similar to those simulated by the training distributions, e.g., we deploy the model to new end users at test time, then we expect that we can effectively adapt to these test distributions for better performance.

in many standard supervised learning setups. For example, we do not access single handwritten characters from an end user, but rather collections of characters such as sentences or paragraphs. When combined with the group assumption above, we arrive at a problem setting that is similar to the standard meta-learning setting (Vinyals et al., 2016). This allows us to extend well established tools and techniques from meta-learning to address distribution shift problems. Meta-learning typically assumes that training data are grouped into tasks and new tasks are encountered at meta-test time, however these new tasks still include labeled examples for adaptation. As illustrated in Figure 1, we instead aim to train a model that uses unlabeled data to adapt to the test distribution, thereby not requiring the model to generalize zero shot to all test distributions as in prior approaches.

The main contribution of this paper is to introduce the framework of adaptive risk minimization (ARM), in which models have the opportunity to adapt to the data distribution at test time based on unlabeled data points. This contribution provides a principled approach for designing meta-learning methods to tackle distribution shift. We introduce an algorithm and instantiate a set of methods for solving ARM that, given a set of candidate distribution shifts, meta-learns a model that is adaptable to these shifts. One such method is based on meta-training a model such that simply updating batch normalization statistics (Ioffe & Szegedy, 2015) provides effective adaptation at test time, and we demonstrate that this simple approach can produce surprisingly strong results. Our experiments demonstrate that the proposed methods, by leveraging the meta-training phase, are able to outperform prior methods for handling distribution shift in image classification settings exhibiting group shift, including benchmarks for federated learning (Caldas et al., 2019) and testing image classifier robustness (Hendrycks & Dietterich, 2019).

## 2 RELATED WORK

A number of prior works have studied distributional shift in various forms (Quiñonero Candela et al., 2009). In this section, we review prior work in robust optimization, meta-learning, and adaptation.

**Robust optimization.** DRO methods optimize machine learning systems to be robust to adversarial data distributions, thus optimizing for worst case performance against distribution shift (Globerson & Roweis, 2006; Ben-Tal et al., 2013; Liu & Ziebart, 2014; Esfahani & Kuhn, 2015; Miyato et al., 2015; Duchi et al., 2016; Blanchet et al., 2016). Recent work has shown that these algorithms can be utilized with deep neural networks, with additional care taken for regularization and model capacity (Sagawa et al., 2020). Unlike DRO methods, ARM methods do not require the model to generalize zero shot to all test time distribution shifts, but instead trains it to adapt to these shifts.

Also of particular interest are methods for robustness or adaptation to different users (Horiguchi et al., 2018; Chen et al., 2018; Jiang et al., 2019; Fallah et al., 2020; Lin et al., 2020), a setting commonly referred to as robust or fair federated learning (McMahan et al., 2017; Mohri et al., 2019; Li et al., 2020). Unlike these works, we consider the federated learning problem setting in which we do not assume access to any labels from any test users, as we partition users into disjoint train and test sets. We argue that this is a realistic setting for many practical machine learning systems – oftentimes, the only available information from the end user is an unlabeled batch of data.

**Meta-learning.** Meta-learning (Schmidhuber, 1987; Bengio et al., 1992; Thrun & Pratt, 1998; Hochreiter et al., 2001) has been most extensively studied in the context of few shot supervised learning methods (Santoro et al., 2016; Vinyals et al., 2016; Ravi & Larochelle, 2017; Finn et al., 2017; Snell et al., 2017), i.e., *labeled* adaptation. The aim of this work is to extend meta-learning paradigms to problems requiring unlabeled adaptation, with the goal of tackling distribution shift. We demonstrate in the next section how paradigms such as contextual meta-learning (Garnelo et al., 2018; Requeima et al., 2019) are readily extended using the ARM framework.

Some other meta-learning methods adapt using both labeled and unlabeled data, either in the semi supervised learning setting (Ren et al., 2018; Zhang et al., 2018; Li et al., 2019) or the transductive learning setting (Liu et al., 2019; Antoniou & Storkey, 2019; Hu et al., 2020). These works do not focus on the same setting of distribution shift and all assume access to labeled data for adaptation. Prior works in meta-learning for unlabeled adaptation include Yu et al. (2018), which adapts a policy to imitate human demonstrations in the context of robotic learning, and Metz et al. (2019), which meta-learns an update rule for unsupervised representation learning, though they still require labels to learn a predictive model. Unlike these prior works, the ARM framework facilitates the development of meta-learning methods for quickly adapting a predictive model using unlabeled examples.

**Adaptation to shift.** Unlabeled adaptation has primarily been studied separately from meta-learning. Domain adaptation is a prominent framework that assumes access to test examples at training time, similar to transductive learning (Vapnik, 1998). Some of these methods, such as importance weighting approaches (Shimodaira, 2000), only handle a single predefined shift and do not constitute test time adaptation (Csurka, 2017; Wilson & Cook, 2020). Certain domain adaptation methods, however, are applicable in the setting with training groups, such as methods for learning invariant features (Ganin & Lempitsky, 2015; Li et al., 2018), and we compare to these methods in Section 4. Several methods for adaptation at test time have been developed specifically for dealing with label shift (Royer & Lampert, 2015; Lipton et al., 2018; Sulc & Matas, 2019). Other methods adapt using statistics of the test inputs (Li et al., 2017) or optimize self-supervised surrogate losses (Sun et al., 2020), and these methods have been shown to perform well across a number of image classification domains. We also compare against these prior methods in Section 4.

## 3 Adaptive Risk Minimization

In this section, we first formally describe the ARM problem setting, which builds on the settings used in prior work for tackling distribution shift. The novel aspect of the ARM setting is that it is amenable to meta-learning solutions to shift, and we demonstrate this by proposing an objective for the ARM setting that resembles typical meta-learning objectives. The problem setting and objective together constitute the ARM problem formulation. We subsequently propose a general algorithm as well as specific meta-learning approaches for solving the ARM problem.

### 3.1 The ARM problem setting

A key goal in machine learning is to develop methods that can go beyond the standard ERM setting and generalize in the face of distribution shift. Accomplishing this goal necessitates the use of additional assumptions beyond ERM, and we wish to carefully craft these assumptions such that they fulfill two properties: they are realistic and applicable to real world problems, and they allow for powerful and tractable methods. In this work, we choose two assumptions that are well established in the literature on distribution shift, in order to fulfill the first property, and we develop a novel meta-learning framework using these assumptions, thus fulfilling the second.

The first assumption is that the training data are provided in groups, which, as discussed above, mirrors analogous assumptions made in group DRO (Hu et al., 2018), federated learning (McMahan et al., 2017), and meta-learning (Vinyals et al., 2016), among other settings. The second assumption is that we observe batches of test points all together, rather than one point at a time. Assuming access to multiple test points has been standard in domain adaptation (Csurka, 2017; Wilson & Cook, 2020), which makes this assumption for training, as well as recent works studying test time adaptation (Li et al., 2017; Sun et al., 2020; Wang et al., 2020). To our knowledge, these assumptions have not been considered simultaneously in prior work. However, as we detail in this section, it is their conjunction that allows us to develop meta-learning solutions to shift.

In the ARM problem setting, we assume access to a training dataset that consists of $N$ labeled data points $(\mathbf{x}^{(i)}, y^{(i)}, z^{(i)})$ sampled i.i.d. from the training distribution $p$. As noted, this differs from standard supervised learning in that we additionally observe the group $z^{(i)}$ associated with each point, which is a discrete value $z \in \{1, \ldots, S\}$ that can represent tasks, users, or other types of meta-data. The goal is to learn a model $g(\cdot\,; \theta) : \mathcal{X} \to \mathcal{Y}$ that is parameterized by $\theta \in \Theta$ and predicts the output $y \in \mathcal{Y}$ given the input $\mathbf{x} \in \mathcal{X}$. At test time, we are given batches of $K$ *unlabeled* data points, where each batch is drawn from a distribution that may differ from both $p$ and the other batch distributions, and we do not observe either $y$ or $z$. For example, we can imagine a test scenario that separately considers each user's images, as discussed in Section 1.

## 3.2 Deriving the ARM objective

We approach the goal of learning adaptable models through the lens of meta-learning. In particular, we define an *adaptation model* as a function $h(\cdot\,, \cdot\,; \phi) : \Theta \times \mathcal{X}^K \to \Theta$, which is parameterized by $\phi$. $h$ takes as input the model parameters $\theta$ and $K$ unlabeled data points and produces updated parameters $\theta'$ after adapting using the $K$ points. $h$ can be initialized as a variety of different adaptation procedures, and we defer this discussion to subsection 3.3. Our goal is to meta-learn $\phi$ and $\theta$ such that $h$ can adapt $g$ using *unlabeled* training data sampled according to a particular group $z$. Assuming that we will observe batches of data at test time that exhibit a similar type of shift, we can then perform the same procedure for better test performance. This motivates the ARM objective, given by

$$\min_{\theta, \phi} \mathbb{E}_{p_z} \left[ \mathbb{E}_{p_{\mathbf{x}y|z}} \left[ \frac{1}{K} \sum_{k=1}^{K} \ell(g(\mathbf{x}_k; \theta'), y_k) \right] \right] , \text{ where } \theta' = h(\theta, \mathbf{x}_1, \ldots, \mathbf{x}_K; \phi). \quad (1)$$

A priori, we do not know what $p_{\text{test}}(z)$ will be, i.e., which values of $z$ will be seen at test time. Thus, we draw inspiration from prior work in deep learning that demonstrates that uniformly sampling over a quantity of interest, such as labels or groups, is a strong method for achieving robustness with respect to that quantity (Shen et al., 2016; Buda et al., 2018; Sagawa et al., 2020). We extend this approach to the ARM setting by defining $p(z)$ at training time to place uniform probability mass on each group in the training set, in order to represent all training groups equally. Theoretically, we expect the trained models to perform well at test time when the distribution over *batches* of data matches the training distribution. Note that this is a looser condition than in ERM, as the distribution over individual data points can vary. In practice, similar to how meta-learning for few shot classification has found that meta-learned models can generalize to new meta-test classes (Vinyals et al., 2016; Finn et al., 2017), we empirically show in Section 4 that the trained models can generalize to new test groups.

Standard few shot meta-learning formulations must use disjoint data batches for adaptation and meta-training to avoid label memorization (Vinyals et al., 2016). Since labels are not used during adaptation in ARM, we meta-train the adapted model using the same $K$ examples that are used for adaptation. The labels for these examples are used in the meta-training update but not the adaptation itself. Thus, the adaptation matches the ARM setting at meta-test time, in which $h$ adapts the model on the same unlabeled test points that the adapted model then predicts on.

## 3.3 Optimizing the ARM objective

Algorithm 1 presents a general meta-learning approach for optimizing the ARM objective. In line 5, $h$ outputs updated parameters $\theta'$ using an unlabeled batch of data. We assume that $h$ is differentiable with respect to $\theta$ and $\phi$, thus we can meta-train both $\theta$ and $\phi$ for *post adaptation* performance on a mini batch of data sampled according to a particular group $z$ (line 6). This adaptation is performed using unlabeled data, mimicking the test time procedure detailed in lines 7-8. In practice, we sample mini batches of groups rather than just one group (line 3), to provide a better gradient signal for optimizing $\phi$ and $\theta$.

---

**Algorithm 1** Meta-Learning for ARM

```
// Training procedure
```
**Require:** # training steps $T$, batch size $K$, learning rate $\eta$
1: **Initialize:** $\theta, \phi$
2: **for** $t = 1, \ldots, T$ **do**
3:     Sample $z$ uniformly from training groups
4:     Sample $(\mathbf{x}_k, y_k) \sim p(\cdot\,, \cdot\,|z)$ for $k = 1, \ldots, K$
5:     $\theta' \leftarrow h(\theta, \mathbf{x}_1, \ldots, \mathbf{x}_K; \phi)$
6:     $(\theta, \phi) \leftarrow (\theta, \phi) - \eta \nabla_{(\theta, \phi)} \sum_{k=1}^{K} \ell(g(\mathbf{x}_k; \theta'), y_k)$

```
// Test time adaptation procedure
```
**Require:** $\theta, \phi$, test batch $\mathbf{x}_1, \ldots, \mathbf{x}_K$
7: $\theta' \leftarrow h(\theta, \mathbf{x}_1, \ldots, \mathbf{x}_K; \phi)$
8: $\hat{y}_k \leftarrow g(\mathbf{x}_k; \theta')$ for $k = 1, \ldots, K$

---

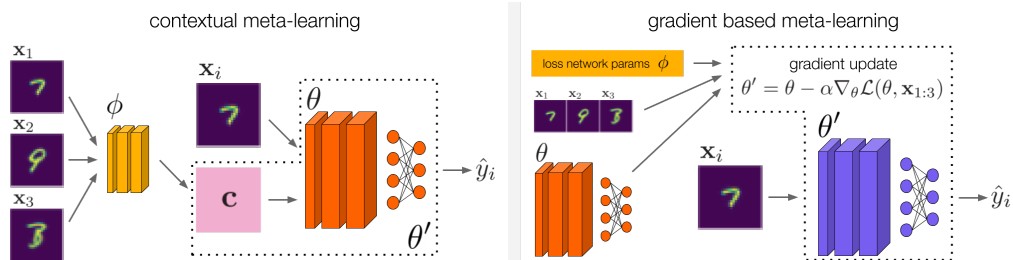

Figure 2: Schematics of the two broad classes of approaches we consider. Left: In the contextual approach, $\mathbf{x}_1, \ldots, \mathbf{x}_K$ are summarized into a context $\mathbf{c}$, and we propose two methods for this summarization, either through a separate context network or using batch normalization activations in the model itself. $\mathbf{c}$ can then be used by the model to infer additional information about the input distribution. Right: In the gradient based approach, an unlabeled loss function $\mathcal{L}$ is used for gradient updates to the model parameters, in order to produce parameters that are specialized to the test inputs and can produce more accurate predictions.

Together, Equation 1 and Algorithm 1 specify how existing meta-learning approaches can be extended to deal with group distribution shift. Most importantly, approaches in which the adaptation model $h$ can be augmented to operate on unlabeled data are readily applicable. We study two such approaches for instantiating the model $g$ and adaptation procedure $h$ in Algorithm 1, which we summarize here and provide full details for in Appendix A. First, we consider a *contextual* approach, shown in Figure 2 (left), in which $h$ summarizes the inputs $\mathbf{x}_1, \ldots, \mathbf{x}_K$ into a context $\mathbf{c}$, which is then used by $g$ as an additional input for predicting on each test point. In this setup, $h$ can learn to provide useful information about the entire batch of $K$ unlabeled data points to $g$ for predicting the correct outputs. In the ARM setup, $g$ is only ever evaluated after adaptation, i.e., with $\theta'$. We can view $h$ as outputting a concatenation of the model parameters and the context $\theta' = [\theta, \mathbf{c}]$.

This approach is inspired by recent works in contextual meta-learning with deep neural networks (Garnelo et al., 2018; Requeima et al., 2019). In line with these works, we propose an ARM-CML implementation of this approach which meta-learns a *context network* $f_{\text{cont}}(\cdot\,; \phi) : \mathcal{X} \to \mathbb{R}^D$. Note that $f_{\text{cont}}$ is parameterized by $\phi$, the parameters of $h$, as in this method, $h$ has no additional parameters. $f_{\text{cont}}$ processes each example $\mathbf{x}_k$ in the mini batch separately to produce $\mathbf{c}_k \in \mathbb{R}^D$ for $k = 1, \ldots, K$, where $D$ is a hyperparameter. The average $\mathbf{c} = \frac{1}{K} \sum_{k=1}^{K} \mathbf{c}_k$ is then used as the context.

Prior works outside of meta-learning have also investigated ways of conditioning predictions on a batch of data. One prominent technique, assuming that the model $g$ is parameterized by a deep neural network with batch normalization layers (Ioffe & Szegedy, 2015), is to compute the normalization statistics for these layers using batches of test inputs, rather than the standard test time procedure of using the running statistics computed over the course of training. Several works have demonstrated the empirical effectiveness of this simple strategy, e.g., Li et al. (2017); Kaku et al. (2020); Nado et al. (2020); Schneider et al. (2020). One advantage of this method's simplicity is that it is easy to translate into the ARM setting, in order to arrive at a meta-learning version of this method which we call ARM-BN. The key difference is that, in ARM-BN, the model is trained to adapt using batches of training points sampled from the same group, following Algorithm 1. We can interpret this method through the contextual approach described above: if we view the running statistics used by standard BN as learned parameters of the model, then $h$ replaces these parameters with statistics computed on the batch of inputs, which then serves as the context $\mathbf{c}$. In ARM-BN, the model is meta-trained to make effective use of this adaptation procedure, thus leading to more effective adaptation at test time. We provide complete details on ARM-BN in Appendix A.

As shown in Figure 2 (right), a distinct approach draws inspiration from gradient based meta-learning, where the goal is to learn parameters $\theta$ that are amenable to gradient updates on a loss function in order to quickly adapt to a new problem (Finn et al., 2017). In other words, $h$ produces $\theta' = \theta - \alpha \nabla_\theta \mathcal{L}(\theta, \mathbf{x}_1, \ldots, \mathbf{x}_K)$, where $\alpha$ is a hyperparameter. Note that the loss function $\mathcal{L}$ used in the gradient updates may be different from the original supervised loss function $\ell$. In particular, in the setting of unlabeled adaptation, $\mathcal{L}$ must be defined such that it operates on only the inputs $\mathbf{x}$, rather than the input output pairs that $\ell$ receives. Akin to Yu et al. (2018), we propose a learned loss (ARM-LL) method that learns to modulate the output features of the model $g$. In our implementation, we assume that $g$ produces output features $\mathbf{o} \in \mathbb{R}^{|\mathcal{Y}|}$ that are used as logits when making predictions. With this

assumption, we define $\mathcal{L}$ to be the composition of $g$ and a *loss network* $f_{\text{loss}}(\cdot\,; \phi) : \mathbb{R}^{|\mathcal{Y}|} \to \mathbb{R}$, which takes in the output features from $g$ and produces a scalar. Note that, similar to the CML method, $f_{\text{loss}}$ is parameterized by $\phi$, as $h$ has no additional parameters. The $\ell_2$-norm of these scalars across the batch of test inputs is used as the loss for updating the model parameter $\theta$. In other words,

$$\mathcal{L}(\theta, \mathbf{x}_1, \ldots, \mathbf{x}_K) = \|v\|_2 \,, \text{ where } v = [f_{\text{loss}}(g(\mathbf{x}_1; \theta); \phi), \ldots, f_{\text{loss}}(g(\mathbf{x}_K; \theta); \phi)] \,.$$

Before adaptation, the output features $\mathbf{o}$ from $g$ need not be suitable logits for prediction, as $g$ is not evaluated for predictive performance using the unadapted parameters $\theta$. Instead, $\mathbf{o}$ may represent, for example, general features of the input $\mathbf{x}$. These features can then be used by $f_{\text{loss}}$ in order to provide a gradient signal that adapts $g$ to output accurate logits.

## 4 EXPERIMENTS

Our experiments are designed to answer the following questions:

(1) Do methods for adaptive risk minimization learn models that can adapt to group shift?

(2) How do these methods compare to prior methods for robustness, invariance, and adaptation?

(3) Can we loosen the assumptions of accessing groups, at training time, and batches, at test time?

### 4.1 EVALUATION DOMAINS AND PROTOCOL

We evaluate on four image classification benchmarks, which span a range of problem settings including federated learning and robustness, demonstrating the general applicability of the proposed methods. Experimental details are provided in full in Appendix B.

**Rotated MNIST.** We study a modified version of MNIST where images are rotated in 10 degree increments, from 0 to 130 degrees. We use only 108 training data points for each of the 2 smallest groups (120 and 130 degrees), and 324 points each for rotations 90 to 110, whereas the overall training set contains 32292 points. At test time, we generate images from the MNIST test set with a certain rotation, and we consider each method's worst case and average accuracy across groups.

**Federated Extended MNIST (FEMNIST).** The extended MNIST (EMNIST) dataset (Cohen et al., 2017) consists of images of handwritten uppercase and lowercase letters, in addition to digits. FEMNIST (Caldas et al., 2019) is a version of EMNIST that associates each handwritten character with its user. We measure each method's worst case and average accuracy across 35 test users, which are held out and thus *disjoint* from the training users.

**Corrupted image datasets.** We evaluate the proposed methods and all comparisons on modified versions of CIFAR-10-C and Tiny ImageNet-C (Hendrycks & Dietterich, 2019), which augment the CIFAR-10 (Krizhevsky, 2009) and Tiny ImageNet datasets, respectively, with common image corruptions that vary in type and severity. We modify the protocol from Hendrycks & Dietterich (2019) to fit into the ARM problem setting by using a set of 56 corruptions for the training data, and we define each corruption to be a group. We use a disjoint set of 22 corruptions for the test data, and we measure worst case and average accuracy across the test corruptions.

### 4.2 COMPARISONS AND ABLATIONS

We compare the ARM methods against several prior methods designed for robustness and adaptation. We summarize the comparisons here and again provide additional details in Appendix B.

**Group robustness and invariance.** Sagawa et al. (2020) recently proposed a state-of-the-art method for group robustness, and we refer to this approach as distributionally robust neural networks (DRNN). Their work also evaluates a strong upweighting (UW) baseline that samples uniformly from each group, and so we also evaluate this approach in our experiments. Note that, for CIFAR-10-C and Tiny ImageNet-C, UW is equivalent to ERM, as the groups all have an equal number of data points. Additionally, we compare to domain adversarial neural networks (DANN) (Ganin & Lempitsky, 2015) and maximum mean discrepancy (MMD) feature learning (Li et al., 2018), two state-of-the-art methods for adversarial learning of invariant predictive features.

**Test time adaptation.** We evaluate the general approach of using test batches to compute batch normalization (BN) statistics, which has been proposed in several prior works (Li et al., 2017; Kaku et al.,

| Method | MNIST | | FEMNIST | | CIFAR-10-C | | Tiny ImageNet-C | |
|---|---|---|---|---|---|---|---|---|
| | WC | Avg | WC | Avg | WC | Avg | WC | Avg |
| ERM | $74.3 \pm 1.7$ | $93.6 \pm 0.4$ | $62.9 \pm 1.9$ | $80.1 \pm 0.9$ | $49.6 \pm 0.1$ | $69.8 \pm 0.4$ | $19.3 \pm 0.5$ | $41.4 \pm 0.2$ |
| UW* | $80.2 \pm 0.1$ | $94.8 \pm 0.2$ | $61.8 \pm 0.9$ | $80.1 \pm 0.3$ | — | — | — | — |
| DRNN | $79.3 \pm 1.1$ | $94.8 \pm 0.1$ | $58.1 \pm 0.7$ | $74.4 \pm 0.8$ | $44.5 \pm 0.5$ | $70.7 \pm 0.6$ | $19.9 \pm 0.3$ | $41.6 \pm 0.2$ |
| DANN | $80.3 \pm 1.4$ | $95.1 \pm 0.1$ | $66.5 \pm 0.4$ | $82.3 \pm 0.4$ | $42.4 \pm 0.2$ | $69.1 \pm 0.4$ | $20.5 \pm 0.1$ | $41.9 \pm 0.2$ |
| MMD | $82.8 \pm 2.0$ | $95.6 \pm 0.3$ | $65.3 \pm 1.6$ | $81.6 \pm 0.6$ | $43.2 \pm 0.4$ | $70.4 \pm 0.3$ | $20.2 \pm 0.0$ | $39.9 \pm 0.5$ |
| BN | $75.1 \pm 0.2$ | $93.9 \pm 0.1$ | $66.9 \pm 0.8$ | $81.1 \pm 0.3$ | $62.5 \pm 0.2$ | $79.4 \pm 0.3$ | $23.9 \pm 0.2$ | $42.8 \pm 0.2$ |
| TTT | $81.1 \pm 0.3$ | $95.4 \pm 0.1$ | $64.1 \pm 0.2$ | $83.4 \pm 0.1$ | $66.6 \pm 0.6$ | $75.6 \pm 0.8$ | $19.7 \pm 0.4$ | $41.4 \pm 0.3$ |
| CML ablation | $78.2 \pm 0.6$ | $94.2 \pm 0.1$ | $64.4 \pm 0.7$ | $81.5 \pm 0.7$ | $47.8 \pm 0.1$ | $68.2 \pm 0.1$ | $19.6 \pm 0.4$ | $42.3 \pm 0.2$ |
| LL ablation | $82.4 \pm 0.3$ | $94.8 \pm 0.2$ | $61.9 \pm 0.2$ | $79.3 \pm 0.6$ | $61.5 \pm 0.2$ | $68.3 \pm 0.5$ | $25.8 \pm 0.4$ | $41.7 \pm 0.1$ |
| ARM-CML | $\mathbf{88.7 \pm 0.6}$ | $\mathbf{96.7 \pm 0.1}$ | $67.8 \pm 1.3$ | $\mathbf{85.7 \pm 0.3}$ | $67.7 \pm 0.5$ | $79.2 \pm 0.3$ | $21.4 \pm 0.2$ | $43.3 \pm 0.4$ |
| ARM-BN | $82.8 \pm 0.4$ | $95.3 \pm 0.1$ | $\mathbf{72.6 \pm 0.3}$ | $\mathbf{85.7 \pm 0.1}$ | $\mathbf{71.1 \pm 0.1}$ | $\mathbf{80.9 \pm 0.2}$ | $\mathbf{27.7 \pm 0.2}$ | $\mathbf{44.9 \pm 0.2}$ |
| ARM-LL | $87.2 \pm 0.5$ | $96.3 \pm 0.2$ | $69.6 \pm 2.1$ | $\mathbf{85.6 \pm 0.5}$ | $66.9 \pm 0.2$ | $75.7 \pm 0.3$ | $27.1 \pm 0.3$ | $44.2 \pm 0.4$ |

Table 1: Worst case (WC) and average (Avg) top 1 accuracy on rotated MNIST, FEMNIST, CIFAR-10-C, and Tiny ImageNet-C across all methods, where means and standard errors are reported across three separate runs of each method. ARM methods consistently achieve greater robustness, measured by WC, and Avg performance compared to prior methods. *The UW baseline is equivalent to ERM for CIFAR-10-C and Tiny ImageNet-C.

2020; Nado et al., 2020; Schneider et al., 2020). We also compare to test time training (TTT) (Sun et al., 2020), which adapts the model at test time using a self-supervised rotation prediction loss. These methods have previously achieved strong results, even without meta-learning, due to their favorable inductive biases for tasks such as image classification (Sun et al., 2020).

Robustness and invariance methods assume access to training groups but not test batches, whereas adaptation methods assume the opposite. Thus, at a high level, we can view the comparisons to these two broad classes of methods as evaluating the importance of each of these assumptions. We also conduct experiments in subsection 4.4 in which we test ARM methods under looser assumptions.

**Ablations.** We also include ablations of the ARM-CML and ARM-LL methods, which sample mini-batches of unlabeled examples uniformly from all groups, rather than sampling from a single group to induce distribution shift. These "context ablation" and "learned loss ablation" are similar to test time adaptation methods in that they do not assume access to training groups. However, these methods lack the inductive bias of BN and TTT, as they instead use learned context and loss networks. These ablations validate the importance of adapting to a specific group.

### 4.3 QUANTITATIVE EVALUATION AND COMPARISONS

In Table 1, we summarize the results. From these results, we highlight several key takeaways:

**ARM methods consistently improve robustness and performance.** Across all of our experiments, ARM methods significantly increase both worst case and average accuracy compared to all other methods. ARM-BN in particular achieves the best performance on most domains, demonstrating the effectiveness of using meta-training to improve an already strong inductive bias that empirically works well for image classification. ARM-CML and ARM-LL also generally improve upon the other methods for almost all metrics, and we suspect that these more expressive methods could perform better than ARM-BN for other modalities such as natural language and video (Yu et al., 2018).

**Robustness methods suffer from pessimism and training difficulties.** DRNN generally results in worse average case and, surprisingly, worst case performance, which we hypothesize may be due to optimization difficulties or overfitting to the training groups. In particular, methods such as DRNN were originally evaluated in settings where the training and test groups were semantically the same (Sagawa et al., 2020), whereas our FEMNIST setup tests on held out users and our CIFAR-10-C and Tiny ImageNet-C setups test on held out corruptions. Indeed, for FEMNIST, we also test $q$-FedAvg (Li et al., 2020), a state-of-the-art method for fair federated learning. $q$-FedAvg was also originally evaluated with the same users at training and test time, and in our setup, this method also performs poorly, achieving $58.2 \pm 1.0$ worst case and $80.8 \pm 0.3$ average accuracy.

We found that, in these experiments, DANN and MMD mostly outperform robustness methods, though the overall performance of these methods for learning invariant features across groups is still worse than ARM methods in general. Invariance methods have primarily been tested in settings with an order of magnitude fewer source domains – usually 4 to 6, rather than tens or hundreds – and more

data per domain (Gulrajani & Lopez-Paz, 2020), and employing techniques such as an adversarial domain classifier may be less effective in general as the number of domains increases.

**With an appropriate inductive bias, test time adaptation methods perform well.** In our evaluation, one of the strongest prior methods is the simple BN method. This method performs well across all metrics, though it typically still lags behind ARM methods and ARM-BN in particular. As discussed above, we believe that this adaptation procedure performs well as it constitutes an inductive bias that is well suited for image classification. TTT offers additional support for this hypothesis: this method also works well across most metrics, in line with previous results on the original versions of the corrupted image benchmarks (Sun et al., 2020), but works notably well for rotated MNIST. This may be because the inductive bias associated with the auxiliary task of rotating images allows the classifier to specifically be more robust to rotation shift.

In summary, in our experiments, we observe poor performance from robustness methods, varying performance from invariance and adaptation methods, and the strongest performance from ARM methods. As ARM methods also make the strongest assumptions, we next present some findings on how these assumptions may be loosened.

### 4.4 LOOSENING THE TRAINING GROUP AND TEST BATCH ASSUMPTION

We present an investigation of the feasibility and effectiveness of ARM methods without the training group and test batch assumptions. Specifically, we consider unsupervised learning techniques for discovering group structure in the training data, as well as a streaming test time evaluation setting.

**Unknown groups.** In the case of unknown groups, one option is to use unsupervised learning techniques to discover group structure in the training data. To test this option, we focus on rotated MNIST and ARM-CML, which performs the best on this dataset, and train a variational autoencoder (VAE) (Kingma & Welling, 2014; Rezende et al., 2014) with discrete latent variables (Jang et al., 2017; Maddison et al., 2017) using the training images and labels. We define the latent variable, which we denote as $c$ to differentiate from the group $z$, to be Categorical with 12 possible discrete values, which we purposefully choose to be smaller than the number of rotations. The VAE is not given any information about the ground truth $z$; however, we encode the notion that $c$ is independent of $y$ by conditioning the decoder on the label. We use the VAE inference network to assign groups to the training data, and we run ARM-CML using these learned groups. In Table 2, we see that ARM-CML in this setting outperforms ERM and is competitive with TTT, which as discussed earlier encodes a strong inductive bias for solving this task. Figure 3 visualizes samples from the VAE for different values of $y$ and $c$.

| Method | WC | Avg |
|---|---|---|
| ERM | $74.3 \pm 1.7$ | $93.6 \pm 0.4$ |
| TTT | $\mathbf{81.1 \pm 0.3}$ | $\mathbf{95.4 \pm 0.1}$ |
| ARM-CML | $\mathbf{81.7 \pm 0.3}$ | $95.2 \pm 0.3$ |

Table 2: Using learned groups, ARM-CML outperforms ERM and matches the performance of TTT on rotated MNIST. This result may be improved by techniques for learning more diverse groups for meta-training.

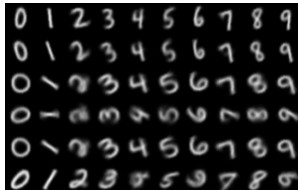

Figure 3: Visualizing VAE samples conditioned on different values of $y$ (x axis) and $c$ (y axis). The VAE learns to use $c$ to represent rotations.

This result suggests that, when group information is not provided, a viable approach is to learn groups for ARM methods. Discovering disentangled factors of variation without supervision is, in the most general sense, an impossible problem (Locatello et al., 2019). However, when combined with meta-learning, the learned groups need not perfectly reflect the test time distributions; rather, the groups should cover many different distributions to allow for meta-training the model such that it can adapt to new test distributions. This advantage was noted by Hsu et al. (2019), who show that even simple techniques such as overcomplete clustering can be effective for defining meta-training tasks. Incorporating techniques from this prior work is a promising direction for building on our results.

**Streaming test points.** When we cannot access a batch of test points all at once, we can augment the proposed ARM methods to be sequential. For example, ARM-CML and ARM-BN can update their average context and normalization statistics, respectively, after observing each new test point. In Figure 4, we visualize the performance of both of these methods, using models that were meta-trained with batch sizes of 50 but evaluated in this streaming setting. We see that both ARM-CML and ARM-BN are able to achieve near their original worst case and average accuracy within observing 10 data points, for rotated MNIST, and 25 data points, for Tiny ImageNet-C, well before the training

Figure 4: In the streaming setting, ARM methods reach strong performance on rotated MNIST (left) and Tiny ImageNet-C (right), after fewer than 10 and 25 data points, respectively, despite meta-training with batch sizes of 50 for both domains. This highlights the ability of the trained models to adapt with small test batches.

batch size of 50. We describe in detail how each ARM method can be applied to the streaming setting in Appendix A. Next, we qualitatively analyze why ARM-CML achieves better performance compared to ERM in the case of FEMNIST.

### 4.5 QUALITATIVE ANALYSIS AND OBSERVATIONS

In Figure 5, we present an example of how ARM methods can improve test accuracy by adapting to specific users. We visualize a batch of 50 examples from a random FEM-NIST test user, and we highlight an ambiguous example. An ERM trained model and an ARM-CML trained model, when only given a test batch size of 2 as shown by the black dashed box, incorrectly classify this example as "2". However, when given access to the entire batch of 50 images, which contain examples of class "2" and "a" from this user, the ARM-CML trained model successfully adapts its prediction to "a", which is the correct label. In general, we find that most examples of adaptation in FEM-NIST occur for similarly ambiguous examples, e.g., "l" versus "I", though not all examples were interpretable.

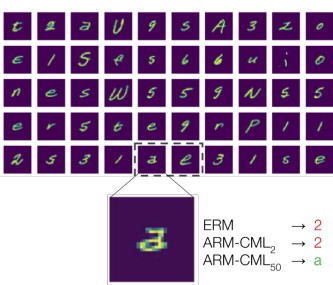

Figure 5: Visualizing one batch of 50 images from a FEMNIST test user. The ARM-CML model, using the entire batch, is able to successfully adapt to output the correct label "a" on the ambiguous example, shown enlarged, whereas other models incorrectly output "2".

## 5 DISCUSSION AND FUTURE WORK

We presented adaptive risk minimization (ARM), a problem formulation for learning models that can robustly adapt in the face of group distribution shift at test time using only a batch of unlabeled test examples. We devised an algorithm and a set of methods for optimizing the ARM objective that meta-learns models that are adaptable to different distributions of training data. Empirically, we observed that ARM methods consistently improve performance in terms of both average and worst case metrics, as compared to a number of prior approaches for handling shift. Two exciting directions for future work are to further explore the unknown groups setting, potentially drawing inspiration from Hsu et al. (2019) as discussed, and to develop more sophisticated ARM approaches.

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

## A  DETAILED DESCRIPTIONS OF META-LEARNING APPROACHES

For ARM-CML, we introduce two neural networks: a context network $f_{\text{cont}}(\cdot\,;\phi) : \mathcal{X} \to \mathbb{R}^D$, as mentioned in subsection 3.3, and a *prediction network* $f_{\text{pred}}(\cdot,\cdot\,;\theta) : \mathcal{X} \times \mathbb{R}^D \to \mathcal{Y}$, parameterized by $\theta$. As discussed, $f_{\text{cont}}$ processes each example $\mathbf{x}_k$ in the mini batch separately to produce contexts $\mathbf{c}_k \in \mathbb{R}^D$ for $k = 1, \ldots, K$, which are averaged together into $\mathbf{c} = \frac{1}{K} \sum_{k=1}^{K} \mathbf{c}_k$. In our experiments, we choose $D$ to be the dimensionality of $\mathbf{x}$, such that we can concatenate each $\mathbf{x}_k$ and $\mathbf{c}$ along the channel dimension to produce the input to $f_{\text{pred}}$. Thus, $f_{\text{pred}}$ processes each $\mathbf{x}_k$ separately to produce an estimate of the output $\hat{y}_k$, but it additionally receives $\mathbf{c}$ as input. In this way, $f_{\text{cont}}$ can provide information about the entire batch of $K$ unlabeled data points to $f_{\text{pred}}$ for predicting the correct outputs.

Figure 6: During inference for ARM-CML, the context network produces a vector $\mathbf{c}_k$ for each input image $\mathbf{x}_k$ in the batch, and the average of these vectors is used as the context $\mathbf{c}$ is input to the prediction network. This context may adapt the model by providing helpful information about the underlying test distribution, and this adaptation can aid prediction for difficult or ambiguous examples. During training, we compute the loss of the post adaptation predictions and backpropagate through the inference procedure to update the model.

A schematic of this method is presented in Figure 6. The post adaptation model parameters $\theta'$ are $[\theta, \bar{c}]$. Since we only ever use the model after adaptation, both during training and at test time, we can simply define $g(\mathbf{x}; \theta') = f_{\text{pred}}(\mathbf{x}, \mathbf{c}; \theta)$, leaving the model's behavior before adaptation undefined. We then also see that $h$ is a function that takes in $(\theta, \mathbf{x}_1, \ldots, \mathbf{x}_K)$ and produces $\left[\theta, \frac{1}{K} \sum_{k=1}^{K} f_{\text{cont}}(\mathbf{x}_k; \phi)\right]$. In the streaming setting tested in subsection 4.4, we keep track of the average context over the previous test points $\mathbf{c}$ and we maintain a counter $t$ of the number of test points seen so far.[1] When we observe a new point $\mathbf{x}$, we increment the counter and update the average context as $\frac{t}{t+1}\mathbf{c} + \frac{1}{t+1} f_{\text{cont}}(\mathbf{x}; \phi)$, and then we make a prediction on $\mathbf{x}$ using this updated context. Notice that, with this procedure, we do not need to store any test points after they are observed, and this procedure results in an equivalent context to ARM-CML in the batch test setting after observing $K$ data points.

At a high level, ARM-BN operates in a similar fashion to ARM-CML, thus we group these methods together into the umbrella of contextual approaches. However, most of the details are different. For ARM-BN, there is no context network, and $h$ has no parameters, i.e., $\phi$ is empty. The model $g$ is again specified via a prediction network $f_{\text{pred}}$, which must have batch normalization layers. Batch normalization typically tracks a running average of the first and second moments of the activations in these layers, which are then used at test time. Thus, we can view these moments, along with the weights in $f_{\text{pred}}$, as part of $\theta$. ARM-BN instead defines $h$ to swap out these moments for the moments computed via the activations on the test batch. This method is remarkably simple, and in deep learning libraries such as PyTorch (Paszke et al., 2019), the implementation requires the changing of a single line of code. However, as shown in Section 4, this method also performs very well empirically, and it is further boosted by meta-training.

In the streaming setting, ARM-BN is also similar to ARM-CML, however it is slightly more complex due to the requirement of computing second moments. Denote the context after seeing $t$ test points as $\mathbf{c} = [\boldsymbol{\mu}, \boldsymbol{\sigma}^2]$, the mean and variance of the batch normalization layer activations on the points so far. Upon seeing a new test point, let $\mathbf{a}$ denote the batch normalization layer activations computed from this new point, with size $h$. We then update the new context to be $\left[\frac{ht}{h(t+1)}\boldsymbol{\mu} + \frac{\sum \mathbf{a}}{h(t+1)}, \frac{ht}{h(t+1)}(\boldsymbol{\sigma}^2 + \boldsymbol{\mu}^2) + \frac{\sum \mathbf{a}^2}{h(t+1)} - (\frac{ht}{h(t+1)}\boldsymbol{\mu} + \frac{\sum \mathbf{a}}{h(t+1)})^2\right]$. Again note that we do not store any test points and that we arrive at the same context as the batch test setting after observing $K$ data points.

---

[1]An alternative to maintaining a counter $t$ is to use an exponential moving average, though we do not experiment with this option.

Finally, for ARM-LL, we note that $\phi$ contains only the parameters of the loss network $f_{\text{loss}}$, and $h$ is defined as

$$h(\theta, \mathbf{x}_1, \ldots, \mathbf{x}_K; \phi) = \theta - \alpha \nabla_\theta \|[f_{\text{loss}}(g(\mathbf{x}_1; \theta); \phi), \ldots, f_{\text{loss}}(g(\mathbf{x}_K; \theta); \phi)]\|_2 \,.$$

We found that $\alpha = 0.1$ worked well for our experiments. We used 1 gradient step for both meta-training and meta-testing. Finally, though we did not evaluate ARM-LL in the streaming setting, in principle this method can be extended to this setting by performing a single gradient step with a smaller $\alpha$ after observing each test point. In an online fashion, we can continually update the model parameters over the course of testing rather than initializing from the meta-learned parameters for each test point.

## B  ADDITIONAL EXPERIMENTAL DETAILS

When reporting our results, we run each method across three seeds and report the mean and standard error across seeds. Standard error is calculated as the sample standard deviation divided by the square root of 3. We checkpoint models after every epoch of training, and at test time, we evaluate the checkpoint with the best worst case validation accuracy. Training hyperparameters and details for how we evaluate validation and test accuracy are provided for each experimental domain below. All hyperparameter settings were selected in preliminary experiments using validation accuracy only.

We also provide details for how we constructed the training, validation, and test splits for each dataset. These splits were designed without any consideration for the train, validation, and test accuracies of any method. All of these design choices were made either intuitively – such as maintaining the original data splits for MNIST – or randomly – such as which users were selected for which splits in FEMNIST – or with a benign alternate purpose – such as choosing disjoint sets of corruptions.

### B.1  ROTATED MNIST DETAILS

We construct a training set of 32292 data points by replicating 90% of the original training set – separating out a validation set – and then applying random rotations to each image. The rotations are not dependent on the image or label, but certain rotations are sampled much less frequently than others. In particular, rotations of 0 through 20 degrees, inclusive, have 7560 data points each, 30 through 50 degrees have 2160 points each, 60 through 80 have 648, 90 through 110 have 324 each, and 120 to 130 have 108 points each.

We train all models for 200 epochs with mini batch sizes of 50. We use Adam updates with learning rate 0.0001 and weight decay 0.0001. We construct an additional level of mini batching for our method as described in subsection 3.3, such that the batch dimensions of the data mini batches is $6 \times 50$ rather than just 50, and each of the inner mini batches contain examples from the same rotation. We refer to the outer batch dimension as the *meta batch size* and the inner dimension as the batch size. All methods are still trained for the same number of epochs and see the same amount of data. Finally, DRNN uses an additional learning rate hyperparameter for their robust loss, which we set to 0.01 across all experiments (Sagawa et al., 2020).

Due to the large number of groups in this setting, we only compute validation accuracy every 10 epochs. When computing validation accuracy, we estimate accuracy on each rotation by randomly sampling 300 of the held out 6000 original training points and applying the specific rotation, resampling for each validation evaluation. This is effectively the same procedure as the test evaluation, which randomly samples 3000 of the 10000 test points and applies a specific rotation.

We retain the original $28 \times 28 \times 1$ dimensionality for the MNIST images, and we divide inputs by 256. We use convolutional neural networks for all methods with varying depths to account for parameter fairness. For ERM, the UW baseline, and DRNN, the network has four convolution layers with 128 filters of size $5 \times 5$, followed by $4 \times 4$ average pooling, one fully connected layer of size 200, and a linear output layer. Rectified linear unit (ReLU) nonlinearities are used throughout, and batch normalization (Ioffe & Szegedy, 2015) is used for the convolution layers. The first two convolution layers use padding to preserve the input height and width, and the last two convolution layers use $2 \times 2$ max pooling. For our method and context ablation, we remove the first two convolution layers for the prediction network, but we incorporate a context network. The context network uses two convolution layers with 64 filters of size $5 \times 5$, with ReLU nonlinearities, batch normalization, and

padding, followed by a final convolution layer with padding. This last layer has number of filters, of size $5 \times 5$, equal to 12 in the case of MNIST, 3 for CIFAR and Tiny ImageNet-C, and 1 for FEMNIST.

## B.2 FEMNIST DETAILS

FEMNIST, and EMNIST in general, is a significantly more challenging dataset compared to MNIST due to its larger label space (62 compared to 10 classes), label imbalance (almost half of the data points are digits), and inherent ambiguities (e.g., lowercase versus uppercase "o") (Cohen et al., 2017). In processing the FEMNIST dataset,[2] we filter out users with fewer than 100 examples, leaving 262, 50, and 35 unique users and a total of 62732, 8484, and 8439 data points in the training, validation, and test splits, respectively. The smallest users contain 104, 119, and 140 data points, respectively. We keep all hyperparameters the same as MNIST, except we set the meta batch size for our method to be 2.

We additionally compare to $q$-FedAvg on this domain, as this method is specifically designed for federated learning settings (Li et al., 2020). We modify the authors' publicly available code[3] to run experiments in our setting, and we will make this fork available upon publication along with our own code base. This method follows its own update rule and hyperparameter settings, and we separately optimize the hyperparameters for $q$-FedAvg as described in Li et al. (2020). Specifically, we first set $q = 0$ and sweep learning rate values between 0.0001 and 1.0, and then we sweep $q \in \{0.001, 0.01, 0.1, 0.5, 1, 2, 5, 10, 15\}$ with the optimal learning rate. With this procedure, we set learning rate to be 0.8 and $q$ to be 0.001.

We compute validation accuracy every epoch by iterating through the data of each validation user once, and this procedure is the same as test evaluation. Note that all methods will sometimes receive small batch sizes as each user's data size may not be a multiple of 50, and though this may affect ARM methods, we demonstrate in subsection 4.4 that ARM-CML can adapt using batch sizes much smaller than 50. The network architectures are the same as the architectures used for rotated MNIST.

## B.3 CIFAR-10-C AND TINY IMAGENET-C DETAILS

For both CIFAR-10-C and Tiny ImageNet-C, we construct training, validation, and test sets with 56, 17, and 22 groups, respectively. Each group is based on type and severity of corruption. We split groups such that corruptions in the training, validation, and test sets are disjoint. Specifically, the training set consists of Gaussian noise, shot noise, defocus blur, glass blur, zoom blur, snow, frost, brightness, contrast, and pixelate corruptions of all severity levels. Similarly, the validation set consists of speckle noise, Gaussian blur, and saturate corruptions, and the test set consists of impulse noise, motion blur, fog, and elastic transform corruptions of all severity levels. For two corruptions, spatter and JPEG compression, we include lower severities (1-3) in the training set and higher severities (4-5) in the validation and test sets. For the training and validation sets, each group consists of 1000 images for CIFAR-10-C and 2000 images for Tiny ImageNet-C, giving training sets of size 56000 and 112000, respectively. We use the full test set of 10000 images for each group, giving a total of 220000 test images for both CIFAR-10-C and Tiny ImageNet-C.

In these experiments, we train ResNet-50 (He et al., 2016) models with a support size of 50 and meta batch size of 6. As described above, the context ablation and ARM-CML additionally use small convolutional context networks, and the learned loss ablation and ARM-LL use small fully connected loss networks. The images are normalized by the ImageNet mean and standard deviation before they are passed through the model. For CIFAR-10-C, we train models from scratch for 100 epochs, and for Tiny ImageNet-C we fine tune a pretrained model for 50 epochs. We use stochastic gradient descent with learning rate 0.01, momentum 0.9, and weight decay 0.0001. We evaluate validation accuracy after every epoch and perform model selection based on the worst case accuracy over groups. We perform test evaluation by randomly sampling 3000 images from each group and computing worst case and average classification accuracy across groups.

---

[2]https://github.com/TalwalkarLab/leaf/tree/master/data/femnist.
[3]https://github.com/litian96/fair_flearn

