# OpenReview forum: "Adaptive Risk Minimization: A Meta-Learning Approach for Tackling Group Shift"
_ICLR.cc/2021/Conference — Reject_

### Official Review · AnonReviewer3 · 2020-10-21
**Two main concerns to be addressed.**

**Rating:** 5
**Confidence:** 3

**Review:**

Paper summary:

The authors try to tackle the *distribution shift* problem with a meta learning approach. The algorithm, namely ARM, is proposed. Following regular meta learning regime, ARM uses an updated version of parameter $\theta'$ to calculate the loss for back propagation. Several specific implementations are put forward, i.e. the contextual / gradient-based methods. Experiments are performed in small and large scale datasets to demonstrate the effectiveness of the proposed algorithm. Detailed ablation study and qualitative analysis are also conducted.

Comments:

This paper has clear writing and well established methodology. However, the main drawback to me is the lack of novelty. Out-of-distribution generalization and meta learning are both popular areas in the current machine learning community, but a simple combination might not be able to convince me.

A second concern about ARM method is the reliability of the empirical experiments. The [DomainBed](https://github.com/facebookresearch/DomainBed) is a suite which could conduct systematical evaluation for domain shift algorithms. In its [public results](https://github.com/facebookresearch/DomainBed/blob/master/domainbed/results/2020_10_06_7df6f06/results.tex), ARM's performance is consistently worse than ERM (like many other algorithms who claim to improve on OoD setting), in all three validation method (training-domain validation, leave one out, oracle). I am familiar with DomainBed's implementation, so for me this difference indicates that the authors might pick the hyper-parameters and datasets based on testsets, as DomainBed is different from regular evaluation for its random parameter picking and extensive datasets searching. If authors could resolve this concern properly, I will consider raising my score.

---

> ### Author Response · Authors · 2020-11-14
> **Author response to R3**
>
> Thank you for your helpful feedback and comments! We have revised the paper to highlight the main contributions and clarify our experimental protocol. We believe that these changes and the responses below address your key concerns, and please let us know if otherwise.
>
> - “However, the main drawback to me is the lack of novelty.”
>
> The core novel contribution of this work is to formalize how the framework of meta-learning can already provide better solutions to distribution shift problems than current state of the art (SOTA) methods, with the goal of improving the SOTA for tackling distribution shift problems. We have clarified the exposition in Sections 1-3 to better reflect this core contribution. We agree that this contribution seems simple, however, we view the simplicity as a strength, as it provides a framework for extending meta-learning tools to unlabeled adaptation. We demonstrate this by developing ARM-BN, which is a novel meta-learning method for this setting that is simple and effective in practice. Through these contributions, we are able to achieve substantially better empirical results compared to SOTA methods (~1-4% average, ~4-7% worst case) on a range of distribution shift problems.
>
> - “A second concern about ARM method is the reliability of the empirical experiments… If authors could resolve this concern properly, I will consider raising my score.”
>
> Thank you for the opportunity to address this valid concern. First, we note that we definitely do *not*, in any of our experiments, choose hyperparameters, datasets, or anything else based on test results. We now include additional details of our protocol at the top of Appendix B. To summarize briefly, we picked the dataset splits once before running any experiments, and we used only validation performance for hyperparameter selection.
>
> As for the DomainBed results, we believe that, although the domain generalization setting is similar to the ARM setting, there are two critical differences. First, the testbeds in DomainBed are primarily focused on complete OOD generalization, e.g., using photos and sketches to generalize to cartoons. As evidenced by the papers that first introduced these datasets, these problems were often designed to test methods for invariance. This leads to the second key difference, which is that our testbeds feature an order of magnitude more source domains, with less data per domain and more overlap between domains. We believe that this setting presents a different type of distribution shift problem that offers algorithms greater leverage for tackling the shift, while still pertaining to relevant problems (e.g., federated learning). Further, we believe that it is precisely this setting where multiple data points may provide crucial information about the domain (e.g., the user), thus leading to greater adaptation performance. We see evidence of this in the DomainBed results themselves, as ARM is clearly superior in the colored MNIST setting from IRM, where adaptation provides a significant advantage in determining the underlying color correlations. Thus, we believe that both of these types of problems are worth further study, as different types of methods may perform strongly for these problems.
>
> We believe that this should resolve your concern about our empirical results, and again, please let us know if otherwise.

---

> ### Author Response · Authors · 2020-11-19
> **Following up on your score update**
>
> Thank you for increasing your overall score! As it is still borderline, we are wondering whether our previous response fully addressed your concerns and if you have any additional concerns we can address before the end of the rebuttal period. We would appreciate a dialogue in order to both improve the paper and reach a more satisfying consensus.
>
> We have revised the paper according to the comments from the other two reviewers, and we believe that this happened after your score update. One revision that is relevant to your comments from earlier is that we have added comparisons to MMD [Li et al, CVPR ‘18] and DANN [Ganin et al, JMLR ‘16] in Section 4, which are both benchmark methods in DomainBed. Our implementations are based heavily off of the DomainBed implementations, and we tune the same hyperparameters including learning rates and adversarial classifier strength. However, we still observe generally worse performance on our testbeds from these methods compared to ARM methods. We believe that this lends additional evidence to the notion that the ARM setting is different from the typical domain generalization setting -- in practice, different methods will perform well in these settings, which are both realistic and worthy of further study.
>
> We look forward to your response and hope we can engage in a meaningful discussion.

---

### Official Review · AnonReviewer1 · 2020-10-29
**The authors propose adaptive risk minimization (ARM) framework to address the problem of distribution. It is a nice work. A key concern is that the proposed ARM sounds incremental to Meta-Learning approaches.**

**Rating:** 7
**Confidence:** 5

**Review:**

The authors propose adaptive risk minimization (ARM) framework to address the problem of distribution shift using meta-learning approaches.

The main contributions of this work are as follows:

1.	The authors propose ARM by employing meta-learning approaches to solve distribution shift problem.

2.	The authors have done extensive experiments to evaluate their proposed ARM algorithms. The results show that ARM outperforms multiple STOA baselines, including ERM, UW, DRNN, BN, TTT.

I still have a few key concerns below that prevent me from giving an acceptance.

Firstly, the overall contribution seems incremental to meta-learning approaches. The authors highlighted the difference between ARM and ERM as the constraint in eq.(1). However, eq.(1) is pretty much the same as meta-learning objective, with (i) the (task) batch loss is written as a sampling average rather than an expectation; and (ii) the adaptation is done using only x, without labels (y’s). As a result, this works is applying meta-learning paradigm to solve distribution shift problem. The proposed implementations ARM-CML and ARM-LL are black-box and optimization-based meta-learning, respectively. It would be great if the authors could explain in more details of the technical contributions, if I missed anything here.

Second, in Figure 3, the results show that ARM outperforms ERM with different data samples/shots. However, the results were obtained on rotated MNIST, which is a fairly simple task, with the accuracy roughly 96.7\%. It would be nice to show results on more complex tasks, such as Tiny ImageNet-C. Would it be the case that more shots/samples lead to higher accuracy?

________________________

Thanks! The authors' update was well-received. I like the way how ARM formulates the unlabeled adaptation. Though "generalizing" meta-learning from labeled to unlabeled adaptation is natural and simple, ARM (proposed in this work) is crucial to highlight this and present the results to the community. I increased the rating to 7.

---

> ### Author Response · Authors · 2020-11-16
> **Author response to R1**
>
> Thank you for your positive comments and detailed inquiries! We have revised the paper to highlight the main contributions and added the requested experiment in Section 4.4. We believe that these changes and the responses below address your key concerns, please let us know if otherwise.
>
> - “Firstly, the overall contribution seems incremental to meta-learning approaches… It would be great if the authors could explain in more details of the technical contributions, if I missed anything here.”
>
> You are correct in many of the points in this paragraph. However, we do not view this work as being either incremental or not incremental in terms of meta-learning approaches themselves, as the goal is not to propose new meta-learning approaches. The main novel contribution, rather, is the formalization of how the *framework* of meta-learning can already provide better solutions to distribution shift problems than current SOTA methods, with the goal of improving the SOTA for tackling distribution shift problems. We have clarified the exposition in Sections 1-3 to better reflect this core contribution.
>
> Meta-learning has primarily focused on the labeled adaptation setting. Thus the ARM problem formulation *is* a technical contribution, as it provides a framework for extending meta-learning tools to unlabeled adaptation. We demonstrate this by developing ARM-BN, which is a novel meta-learning method for this setting that is simple and effective in practice. Through these contributions, we are able to achieve substantially better empirical results compared to SOTA methods (~1-4% average, ~4-7% worst case) on a range of distribution shift problems.
>
> - “It would be nice to show results on more complex tasks, such as Tiny ImageNet-C. Would it be the case that more shots/samples lead to higher accuracy?”
>
> We now include results in Section 4.4 that test ARM-CML and ARM-BN in this setting on Tiny ImageNet-C. We observe the same trends as for the rotated MNIST experiments, affirming that, in the absence of test batches, ARM methods are still able to adapt quickly to new test settings, often with as few as 25 unlabeled test examples for this domain.

---

> > ### Comment · AnonReviewer1 · 2020-11-23
> > **Though "generalizing" meta-learning from labeled to unlabeled adaptation is natural and simple, ARM (proposed in this work) is crucial to highlight this and present the results to the community.**
> >
> > Thanks! The authors' update was well-received. I like the way how ARM formulates the unlabeled adaptation. Though "generalizing" meta-learning from labeled to unlabeled adaptation is natural and simple, ARM (proposed in this work) is crucial to highlight this and present the results to the community. I increased the rating to 7.

---

> > > ### Author Response · Authors · 2020-11-24
> > > **Thanks!**
> > >
> > > Thanks very much for your response! We appreciate your concise summary of the contributions of this work.

---

### Official Review · AnonReviewer2 · 2020-11-07
**Official Blind Review #2**

**Rating:** 6
**Confidence:** 4

**Review:**

This paper studies domain adaptation under the assumption that only unlabeled target data is available in training and the domain shift follows a special group shift. The main idea for the proposed method is having an adaptation model that takes only the unlabeled data in and output updated parameters. The proposed method also involves the test-time training, which means the adaptation model takes in unlabeled training data in training but takes in unlabeled target data in the adaptation phase. The method is called adaptive risk minimization and there are two meta-learning approaches provided, contextual and gradient-based, in the paper. In the experiment, the proposed method outperforms a limited set of baselines. The paper also discusses a few cases when the assumption is violated like the group indicators are unknown.

Strong points
1. This paper studies an important setting in domain adaptation, similar to unsupervised multisource domain adaptation.
2. The idea of using an adaptation component trained in a meta-learning fashion with test-time adaptation is novel.

Weak points (maybe due to my confusion)


1. Group shift definition is not very intuitive to me, especially how it would affect the proposed method. It seems to be just a different name for “domain”. In the DRO setting [Hu 2018], the group information is used in optimization. However, in this paper, the group information is only used in sampling multiple group data for training and sampling single group data to train ARM-BN. Therefore, the so-called group shift in this paper seems to be a domain shift when there are multiple source domains.

2. It is not clear to me if the test data is not one of the group, or is not the same distribution with one of the training distribution, how the model can be “adaptive”. Before the adaptation step, the model only sees unlabeled training data. In the experiment, especially for rotation mnist, it is not clear whether the test is covered in the training. I think a more principled way to do multiple random separations of rotations/corruptions -- making sure there is no overlap, and then evaluate on the test data.

3. The general setting is very similar to unsupervised multi-source domain adaptation. The difference between domain adaptation to a single fixed target domain with only unlabeled data available with this so-called adaptation to shift setting is very subtle to me. The way to use unlabeled target data for adaptation in the former perspective is very rich, including MMD, importance weighting, adversarial training. More should be discussed in the paper. And many should serve as baselines.

Given the weak points, I recommend weak rejection for this paper.

Additional questions and suggestions:
1. Instead of the group shift assumption, I think a more interesting question is: what is the assumption that will make the proposed method to work? When the model claims to adapt to a new test group that is not in the training, usually you assume the group/domain does not change the label. This seems to be the case from the data used in the paper. Another question is whether y|x is the same between training and testing, this seems to be also the case. So for meta-learning and test time training methods, clarifying their assumption is better than just casting a group shift setting to it.

2. The two meta-learning methods, contextual and gradient-based, are not new. I feel they should be discussed more carefully. The novelty of this method is the incorporation of meta-learning-style training and test time adaptation for unlabeled data.

---

> ### Author Response · Authors · 2020-11-16
> **Author response to R2**
>
> Thank you for your insightful comments and feedback! We have revised the paper according to your feedback and added the requested comparisons in Section 4. We believe that these changes and the responses below address your key concerns, please let us know if otherwise.
>
> - “Group shift definition is not very intuitive to me, especially how it would affect the proposed method. It seems to be just a different name for ‘domain’.”
>
> You are correct. Groups can correspond to users, domains, subpopulations etc. We adopt the term following prior work [Sagawa et al, ICLR ‘20], as we believe it is an appropriate term for generally encompassing users, corruptions, and related concepts.
>
> - “It is not clear to me if the test data is not one of the group, or is not the same distribution with one of the training distribution, how the model can be ‘adaptive’.”
> - “Instead of the group shift assumption, I think a more interesting question is: what is the assumption that will make the proposed method to work?”
>
> This is a good point, and we added some discussion for this in Section 3.2. The assumption is similar to what you have suggested, though slightly more general -- we require that the distributions over *batches* **X** and **y|X** match between train and test. Note that the distributions over individual data points **x** and y|**x** can still shift, illustrating the benefit of meta-learning to adapt to multiple unlabeled data points. In practice, possibly due to the generalization capability of neural networks, we find that this assumption only needs to hold loosely. Similar to how typical meta-learning for few shot classification have found that meta-learned models can generalize to new meta-test classes, we find that models trained with ARM methods can generalize to new test groups.
>
> - “In the experiment, especially for rotation mnist, it is not clear whether the test is covered in the training. I think a more principled way to do multiple random separations of rotations/corruptions -- making sure there is no overlap, and then evaluate on the test data.”
>
> Rotated MNIST is set up slightly differently from the other experiments, in that the training and test groups are the same (but not the datapoints in those groups). The challenge is that some groups are heavily underrepresented in the training data, thus models trained with ERM will have poor test accuracy on those groups. The point of this experiment differs from the other three experiments, which as you have suggested, evaluates the situation in which there is no overlap between train and test groups.
>
> - “The general setting is very similar to unsupervised multi-source domain adaptation.”
>
> In the three experiments with no overlap between train and test groups, the primary difference between the ARM setting and domain adaptation is that we do not assume access to any examples, labeled or unlabeled, from the target domain(s) at training time. This complicates the usage of typical domain adaptation methods such as importance weighting.
>
> However, the ARM setting does bear some resemblance to domain generalization, which is perhaps what you are referring to as “multi-source domain adaptation”. Among the methods you pointed out, MMD and adversarial training are also applicable in this setting. Thus, we have added comparisons to MMD [Li et al, CVPR ‘18] and DANN [Ganin et al, JMLR ‘16] in Section 4, and we added some discussion of these methods to Section 2. ARM methods also consistently outperform these methods on all testbeds. Thank you for pointing out this important set of comparisons.
>
> - “The two meta-learning methods, contextual and gradient-based, are not new. I feel they should be discussed more carefully.”
>
> Thank you, we revised Section 3.3 with this in mind. We appreciate your clear summary of the novelty of this work.

---

> > ### Comment · AnonReviewer2 · 2020-11-25
> > **Thanks for the response**
> >
> > I read the response and appreciate the additional baselines. I will update to 6.
> >
> > However, two things that are still unresolved for me: (sorry about my late response, but perhaps these issues deserve more thoughts in general):
> >
> > 1. Group shift concept: in the original group DRO paper, group information is used for providing more structure for the distributionally robust constraint sets, by "grouping" data together in the training data. However, in the data/domain/distribution shift field, given the terminology is already messy, I'd prefer a more careful usage of the terms, for example with more rigorous definition on the assumptions. (What should be the same and what could be different?)
> >
> > 2. Related to the end of 1, the requirement that the distribution of batches of X and Y|X should be the same between source and the target is still vague (even maybe wrong, or ill-defined). Do you mean you regard batches of X as a big vector concatenated by all the X? And when this would be satisfied?

---

### Author Response · Authors · 2020-11-20
**Following up with all reviewers**

To all reviewers, we believe that our previous responses fully addressed your concerns, but we are happy to provide further revisions and experiments should you have any additional concerns. As the window for discussion is closing soon, we are hoping to engage with you in case there are questions and comments we can address before the end of the rebuttal period.

---

### Author Response · Authors · 2020-11-22
**Hoping to receive responses and updated reviews**

We are still hoping that the reviewers will acknowledge our responses to your reviews and update your scores accordingly. There are only two days left in the rebuttal period, so hopefully the reviewers are satisfied by the experiments we have already provided in response to the points you have brought up. But we are still able to answer any outstanding questions and provide revisions to the paper, so we would appreciate some dialogue as to whether you have any additional concerns.

---

### Decision · Program_Chairs · 2021-01-07
**Final Decision**

**Decision:**

Reject

**Comment:**

Dear Authors,

Thank you very much for your detailed feedback to the reviewers in the rebuttal phase. The feedback certainly clarified some of the concerns raised by the reviewers and improved their understanding of your work. Indeed, some of the reviewers have increased their scores.

However, overall, we think this paper has rather marginal novelty and there are still several conceptual and technical issues to be further discussed, such as the definition of the grouping concept and the distributional shift assumption.

For these reasons, I suggest rejection of this paper, in comparison with many other strong submissions. I hope that the detailed feedback and additional comments from the reviewers help you improve this work for future publication.